# The Effects of Modified Graded Recession, Anteriorization and Myectomy of Inferior Oblique Muscles on Superior Oblique Muscle Palsy

**DOI:** 10.3390/jcm10194433

**Published:** 2021-09-27

**Authors:** Yu-Te Huang, Jamie Jiin-Yi Chen, Ming-Yen Wu, Peng-Tai Tien, Yung-Ping Tsui, Yi-Ching Hsieh, Hui-Ju Lin, Lei Wan

**Affiliations:** 1Department of Ophthalmology, China Medical University Hospital, China Medical University, Taichung 40447, Taiwan; tonyhuang791112@gmail.com (Y.-T.H.); lilyluu88@gmail.com (J.J.-Y.C.); lilyluu2396@gmail.com (M.-Y.W.); u702054@hotmail.com (P.-T.T.); u601145@yahoo.com.tw (Y.-P.T.); cizica0331@hotmail.com (Y.-C.H.); 2School of Chinese Medicine, College of Chinese Medicine, China Medical University, Taichung 40447, Taiwan; 3Department of Biotechnology, Asia University, Taichung 40402, Taiwan

**Keywords:** excyclotorsion, graded recession and anteriorization, inferior oblique muscle, myectomy, superior oblique palsy

## Abstract

Background: The aim was to investigate the effect of inferior oblique (IO) operation (IO myectomy or graded recession and anteriorization) for unilateral and bilateral superior oblique muscle palsy (SOP); Methods: A total of 167 eyes undergoing IO surgery by a single surgeon between 2008 and 2015 were retrospectively reviewed. The method for treating symmetric bilateral SOP was bilateral IO myectomy (*n* = 102) and the method for treating unilateral SOP or non-symmetric bilateral SOP was IO-graded recession and anteriorization (*n* = 65). Associated clinical results and other factors were analyzed; Results: Head tilt, vertical deviation, IO overaction, SO underaction degree and ocular torsion angle were all clearly changed, but there was no statistically significance between these two procedures. Mean preoperative torsional angle was 15.3 ± 6.4 degree, which decreased to 5.3 ± 2.7 degree after surgery. Preoperative torsional angle, IOOA and SOUA degree were all significantly affected in postoperative torsional angle (*p* = 0.025, 0.003 and 0.038). Horizontal rectus muscle and IO muscle operation did not interfere with each other’s results (*p* = 0.98); Conclusions: Symmetric bilateral SOP could be treated with bilateral IO myectomy and IO-graded recession and anteriorization should be reserved for unilateral SOP or non-symmetric bilateral SOP.

## 1. Introduction

Superior oblique muscle palsy (SOP) is one of the most frequent ocular motor abnormalities in ophthalmic practice and is the most common etiology of abnormal head posture in childhood [1,2]. SOP is often accompanied by a head tilt to the contralateral non-paretic side [1,2,3,4]. The palsy is presented as either congenital or acquired. Sometimes, congenital SOP, diagnosed after adulthood as asthenopia and diplopia, cannot be compensated by an abnormal head position [5]. Most of the conditions are recognized as idiopathic palsy. Some idiopathic SOP has a genetic background, as evidenced by the familial occurrence [6]. Besides, Single nucleotide polymorphisms (SNPs) of the genes, expressed in the brain stem trochlear nucleus, have also been detected in some SOP patients [7]. SOP can be unilateral or bilateral. The symptoms of SOP included: head title (measured by the rule like goniometric measurement), vertical deviation (measured by prism), and ocular torsion. 

For a more precise surgical planning, the presence of inferior oblique muscle overaction (IOOA) and superior oblique muscle underaction (SOUA), as well as the degree of deviation in the primary positions, should be measured and considered [2,3,4]. The ocular extorsion and fusional cyclovergence of SOP patients can be measured both subjectively and objectively [8,9]. Subjective torsion measurements include double moddox rod examination, but is not available for uncooperative patients or is incorrect in children [9,10]. Objective measurement methods, including digital fundus photography or direct and indirect opthomoscopy detection, are preferred [10,11]. 

The treatment of SOP is primarily surgical unless the deviation degree is small, and may successfully alleviate the symptoms by prisms [1,2,4]. Surgical treatment of SOP is complex and challenging [1,2,3]. A variety of surgical methods have been proposed to tackle SOP, which often can be accomplished by inferior oblique (IO) weakening (including IO myectomy or IO graded recession and anteriorization), a superior oblique muscle tuck, or additional vertical rectus (contralateral inferior rectus or ipsilateral superior rectus muscle) and horizontal muscle surgery [1,7,12]. Knapp’s treatment scheme of SOP is obeyed by most ophthalmologists [12,13,14]. Nevertheless, the most frequent operation performed on SOP patients is weakening of IO and it is generally accepted that all the symptoms of SOP will improve after IO myectomy and IO-graded recession [11,12,13,14,15]. Reports of efficacy of the two operation methods on SOP patients were rare. In this study, we investigated the effects and symptom relief after the two operation methods in SOP patients.

## 2. Materials and Methods

This retrospective, interventional case series study was conducted at China Medical University Hospital (CMUH) between 2008 and 2015. 

### 2.1. Study Population

Inclusion criteria included patients with unilateral or bilateral SOP undergoing IO myectomy or graded recession and anteriorization, with or without horizontal rectus muscle surgery. The chosen procedure was the standard-of-care at our hospital since 2008. The method for treating symmetric bilateral SOP was bilateral IO myectomy and the method for treating unilateral SOP or non-symmetric bilateral SOP was IO-graded recession and anteriorization. Patients received adjustable sutures; vertical muscle operation (superior oblique, inferior rectus and superior rectus muscles) and other IO muscle operations were excluded from this study. Patients with sensory strabismus or paralytic strabismus other than trochlear nerve palsy were also excluded. Patients with less than 3 months post-operative follow-up or incomplete data were also excluded.

### 2.2. Study Procedure

Comprehensive ophthalmological and ocular motility examinations were performed. Associated clinical factors examined include age at the time of surgery, sex and laterality of the involved eye (unilateral or bilateral) were all reviewed. All subject patients were diagnosed based on clinical examinations consisting of a positive 3-step test and evaluation of extraocular movements. The severity of SOP was defined by the presence of characteristic SOUA, IOOA (documented by subjective scale), head tilt (measured by the rule like goniometric measurement), vertical deviation (measured by prism) and ocular torsion angles (detected by fundus photographs). A further description was as below.

The severity of IOOA was rated between 0 and +4, according to the upper deviation of the pupil in adduction. A score was given for every eyes’ pupil deviation, +1 for 1 mm, +2 for 2 mm, +3 for 3 mm, and +4 for 4 mm of deviation [16,17]. If pupil deviation corresponded to the middle of 2 adjacent grade levels, assigning a subjective rating of +0.5, +1.5, +2.5, or +3.5 was also permitted. The severity of SOUA was rated in a similar way to how IOOA was, as was assessed when the patient was looking inferonasally. Head tilt was detected by a rule-like goniometric measurement with one arm perpendicular to the floor and one arm parallel to the axis across the line of the patient’s nose. Fundus photography was performed and archived with the fundus camera (Topcon, Japan) with or without dilated pupils. Fundus photographs were documented for ocular torsion. At the beginning of each photographic session, the patient’s head was stabilized on the chin rest and the headrest by a well-trained examiner. The patients could not cooperate and obtaining fundus photographs were excluded. As for the torsion, the angle of ocular fundus was measured following the previously suggested method; the angle between a straight horizontal line passing through the optic disc geometric center, and a reference line connecting the macula and the optic disc center (Figure 1a,b), was automatically measured using Adobe Photoshop Extended software CS5 (Version 12.0) (Adobe, San Jose, CA, USA). All data were measured before operation and 3 months after operation.

### 2.3. Surgical Planning and Technique

All operations were done by a single experienced pediatric surgeon (H.J. Lin). Superior oblique exaggerated traction before operation were done in all patients and if the SOP was severe or there was no improvement in SOP after IO procedures noted during operation, an SO tuck was performed in the operation; these patients were excluded from this study. The method for treating symmetric bilateral SOP was bilateral IO myectomy and the method for treating unilateral SOP or non-symmetric bilateral SOP was IO-graded recession and anteriorization. These criteria were followed and became standard at our hospital. Preoperative vertical deviation, head title degree or torsional angle was not considered when choosing a surgical method. If a significant amount of esotropia or exotropia was also present, horizontal rectus muscle surgery was also performed in the same operation.

Operations were performed under local or general anesthesia. In general condition, patients aged under 15 years old received general anesthesia and otherwise received local anesthesia. A fornix-based conjunctiva approach was performed in patients combined with other extraocular muscle surgeries, and a swan conjunctiva approach was performed when the patients had just received an IO operation in the field after dissecting Tenon’s capsule and isolating the IO muscle. In cases of IO myectomy, 2 hemostatic forceps (WPI LLC., Sarasota, FL, USA) were applied, separated by at least 6 mm and muscle between the hemostats was excised; the residual IO was released again by a muscle hook (ASICO LLC., Westmont, IL, USA) to make sure there was no adhesion. Tenon’s capsule was not sutured in both procedures, but was cleaned to avoid scar and synechia afterwards.

For graded IO recession and anteriorization, the operation methods were similar to those described by Guemes and Lee but in a more simplified version [18,19]. The amount of recession and anteriorization depended on the pre-operative severity of IOOA (+1 to +4) and the surgeon’s experience. IO were detached and reinserted to a location along the temporal border of the inferior rectus muscle. Patients were categorized into four groups based on the inferior/temporal positions of the attachment of the IO muscle (anterior border and posterior border together as one point) with respect to the IR lateral border as follows: +1: 7.0/2.0 mm +2: 5.0/2.0 mm +3: 3.0/0.0 mm and +4: 2.0/0.0 mm (Figure 2). 

### 2.4. Statistical Analysis

Statistical analysis was presented with mean standard deviation. Various clinical factors (age, postoperative change in head tilt, preoperative SOUA, preoperative IOOA, operation method) were analyzed with multivariable linear regression analysis. In the model, the net changes of torsion after operation were used as dependent variables and these clinical factors were selected as independent variables. Pre- and postoperative ocular torsional angle were compared using a paired *t*-test or a Wilcoxon signed-rank test, as appropriate for the normality of the dataset. Changes in vertical deviation and ocular torsional angle were compared using an independent *t*-test. Changes in torsional angle according to Knapp’s classification of SOP were also compared [20]. A *p* value <0.05 was considered clinically significant. All analyses were performed using SPSS statistics software (version 19.0, IBM, Armonk, NY, USA).

## 3. Results

The study group of a total of 102 people with 167 eyes was included. There were 52 male (49%) and 50 female (51%) with an average age of 9.25 ± 0.35y (range 1.5 to 42 y/o) at the time of surgery. Unilateral SOP consisted of 45 eyes, the number of bilateral asymmetrical SOP was 20 eyes and 102 eyes were bilateral asymmetrical SOP. 

As organized in our surgical planning, myectomy was performed in 102 eyes (61.1%) and IO graded recession and anteriorization (Unilateral SOP and bilateral asymmetrical SOP cases) was performed in 65 eyes (38.9 %). Horizontal rectus muscle surgery was concomitantly performed in 117 eyes (70.1%). According to Knapp’s classification, 73 eyes were in class I (greatest hypertropia in opposite up oblique field) and class II (greatest hypertropia in opposite down oblique field), 62 eyes were in class III (greatest hypertropia in entire opposite field), 19 eyes were in class IV (greatest hypertropia in entire opposite field and across the lower field) and class V (greatest hypertropia across lower field) and 9 eyes were class VI (bilateral SOP); 4 eyes were in other classes. 

The clinic data of the patients of pre-operation and post-operation are shown in Table 1 and Table 2. Using a regression model to detect the effects of preoperative torsional angle, IOOA, SOUA degrees revealed these variables to significantly affect net changes of torsion after operation (regression model *p* < 0.001, coefficient = 0.025, 0.003, and 0.038, respectively) (Table 1).

After surgery, vertical deviation, IOOA, SOUA and torsional angle were all evidently decreased in all patients in the two operation methods (IO myectomy or graded recession and anteriorization) and there were no significant changes between the two operation procedures according to our surgical planning protocols, (*p* = 0.94, 0.89, 0.83, 0.93 and 0.98, respectively) (Table 1). There were also no significant differences in age at surgery (*p* = 0.93) (Table 1). A the combination of horizontal rectus muscle surgery would not change the efficacy of both surgical procedures (*p* = 0.98) (Table 2). Besides, since ocular torsion is a very important, etiology induces a head tilt in SOP patients and the preoperative and postoperative ocular torsion data were summarized separately in Table 2. Mean preoperative torsional angle was 16.3 ± 6.4 degrees, which decreased to 5.3 ± 2.7 degrees after surgery. The two IO surgical procedures with no significant differences influenced post operation ocular torsion (*p* = 0.976) (Table 2).

According to the postoperative correction condition, the patients were divided into 2 groups (Table 3). The definition of full correction was residual vertical deviation (<4 prism), IOOA (<+2), and SOUA (<+2)) (group 1). The definition of partial correction was either one of these signs presented (residual vertical deviation > 4 prism, IOOA > +2 or SOUA > +2) (group 2). Postoperative torsional changes were not significant between these two groups (*p* = 0.064), although the torsion changes were mildly larger in full correction groups (10.18 ± 4.2 vs. 9.73 ± 3.5) (Table 3). The torsional change was also without significant effects by the amount of pre-operation vertical deviation correction (*p* = 0.25) (Table 3). Nevertheless, there was a significant difference between the changes in torsional angle among the subgroups of Knapp’s classification (*p* = 0.023 as compare Knapp’s classification I–III with Knapp’s classification IV–VI) (Table 3). No acute ocular complications were encountered in any patients and no cases were found to have upgaze limitation and hypertropia on the opposite side at 3 months after operation.

## 4. Discussion

SOP is one of the most common head title etiologies in children and various diagnostic and surgical techniques have been introduced to evaluate and correct it. An inferior oblique weakening procedure has been widely accepted as the primary procedure due to secondary inferior oblique overaction which widely accompanies it [21,22]. The surgical methods to weaken overacting inferior oblique muscles are diverse and include myectomy, anteriorization, graded recession and anteriorization [1,7,12].

Both IO myectomy and IO-graded recession and anteriorization had different advantages. The strengths of surgical myectomy are its simplicity, its timesaving nature and its symmetry. However, some studies suggested that the ends of the broken muscle might re-adhere to the sclera and could result in the recurrence of IOOA [23]. Moreover, the effects might be less prominent than in IO-graded recession and anteriorization in patients with dissociated vertical deviation [24]. In our experience, without grading, and just by using IO myectomy in all cases of SOP and IOOA, post operation reversed vertical deviation is often induced, especially when combined with horizontal exotropia rectus muscles. This might be due to a high ratio of masked unilateral SOP and post-operation muscle, tendon and wound synechia. Despite these disadvantages, we still consider this procedure to play an important role in IO weakening.

Many studies have proposed the benefits of IO-graded recession and anteriorization [18,19]. In our experience, IO recession and anteriorization had a better efficacy than IO myectomy in unilateral SOP and non-symmetric bilateral SOP patients. However, the procedure was more challenging than myectomy and is considered a time-consuming operation. Furthermore, the amount of recession and anteriorization varies in different studies and is often solely based on operators’ experience, which made standardization difficult. Guemes and Wrigh reported the recession of the IO muscle 1-, 2-, and 3-mm posterior recessions, with average corrections of 20, 18, and 15 PD, respectively [18]. Lee further divided the procedure into 6 grades, with average reductions of 21.0 to 6.8 PD [19]. Our surgical planning was similar to Lee’s study and a simpler four categories were used. The results were also comparable with the previous studies.

In our study, the rationale for choosing an IO-weakening procedure was as follows. In the cases of bilateral symmetrical SOP, a bilateral IO myectomy method was a simpler and more effective method than graded recession and anteriorization of IO. Bilateral IO-graded recession and anteriorization in this condition was more complex and might have induced an unexpected vertical deviation in symmetric SOP cases. This could be because completely perfect symmetric-graded recession was difficult in both eyes and the intrinsic asymmetrical post operation scar formation around the muscles could also be the predisposition. 

Some studies have examined changes in ocular torsion changed after SOP operation and some studies have reported that an SO tuck was more effective in decreasing subjective and objective extorsion as compared with IO anterior transposition [15]. In this study, IO myectomy or graded recession were performed in SOP patients and we found that the two procedures were all effective in improving ocular torsion. We cannot deny that SO tuck is absolutely needed in severe SOP cases and SO exaggerated traction test before and during operation may help us to decide that an SO tuck is needed in the same operation.

In a number of studies, single muscle surgery has been suggested to be inadequate for patients who presented with large hypertropias (mostly defined as >15 PD) in primary gaze [21,22,25]. An SO tightening procedure was advocated in this situation. However, in our studies, the average amount both groups treated either with IO myectomy and graded recession and anteriorization were larger than 15 PD (18 and 20 PD). Great results could still be achieved under refined surgical planning, and a prudent and delicate operation to avoid scar formation and synechia.

As for cyclodeviation, the mean torsional angle was 15.3 degrees before surgery and 5.3 degrees after operation. Previous reports have revealed that postoperative ocular torsion reduction may be temporary and it would begin a decline at 10 weeks after surgery [15]. However, our fundus photographs were taken 3 months after surgery, and the average following up period was 3.5 years. Therefore, the torsion improvement after operation is long-standing in our study, even when we used 5-0 dexon absorbable sutures in our surgeries. In cases of combining horizontal rectus muscle, all patients’ postoperative ocular torsion was significantly improved compared with preoperative values in both groups, indicating that horizontal rectus muscle surgery does not influence postoperative changes in ocular torsion. The results do not interfere each other and this result is comparable with other studies [22]. Consequently, we suggest that horizontal rectus muscle surgeries and vertical strabismus surgeries might not be needed to be divided into separated procedures. 

Knapp’s classical classification criteria for SOP were well used in designing operation procedures and the seven-grade division had been considered as the basis for classification and treatment of the condition to date [20]. In our studies, the decrease in torsion was more significant in Knapp’s classification IV–VI than in Knapp’s classification I–III (*p* = 0.023). This might be due to the intensity of the SOP being higher in Knapp’s classification IV–VI, so the changes were also higher after operation. Nevertheless, they all had a significant change afterwards in all classes after IO procedures in our study. 

In a linear regression model analysis, the data revealed that the effect of various clinical preoperative factors—preoperative torsional angle, IOOA and SOUA—all significantly influenced postoperative ocular torsion. Similar results were also proposed in other studies [26,27]. It was not surprising that these symptoms were all closely related to ocular torsion, and the changes after operation were also “yoked together”. However, the effects of preoperative IOOA and SOUA were not to have the same effects (*p* = 0.025 and 0.003, respectively). Consequently, the amount of SOUA and IOOA could not give an equation and must be checked individually and be taken into consideration in operative designs.

Several limitations remain in the current study and most of the biases come from the retrospective nature of this study. Second, since these two procedures had different indications, there was no direct comparison between both procedures for similar cases. However, we could conclude that our treatment scheme was effective and efficient. Moreover, the small sample size and short follow-up for determining risk of anti-elevation were also a concern. We could not judge the long-term effects of these two procedures. However, several studies focusing on the surgical outcomes of SOP showed similar results [15,22,25,26,27]. Nevertheless, further large-scale prospective studies should be conducted to fully evaluate treatment efficacy and to establish the optimal treatment strategy of superior oblique muscle palsy.

## 5. Conclusions

IO weakening procedures were effective and safe when treating SOP. We propose that symmetric bilateral SOP should be treated with bilateral IO myectomy and the method for treating unilateral SOP or non-symmetric bilateral SOP was IO graded recession and anteriorization. Each procedure can achieve excellent and predictable results in objective ocular torsion, cyclodeviation and vertical deviation. Combining horizontal muscles in IO operations was found to be safe and did not interrupt either operation. 

## Figures and Tables

**Figure 1 jcm-10-04433-f001:**
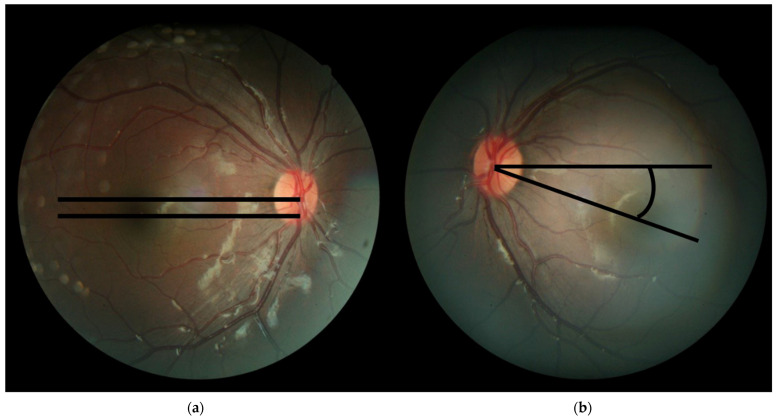
Assessment of objective ocular torsion on fundus photography. (**a**) D The cyclotorsion angle was measured from two straight lines: the first line passing through the of the geometric center of optic disc and the second line joining the center of the optic disc to the fovea; (**b**) Eye fundus with excyclotorsion: Extorsion was defined by the fovea located below the lower edge of the optic disc. Angle is the degree between the straight line passing through the optic disc connecting the macula and the reference line.

**Figure 2 jcm-10-04433-f002:**
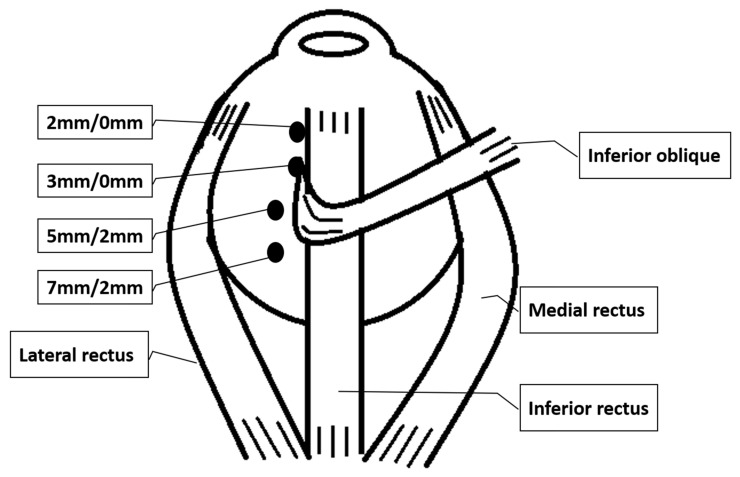
Surgical planning for IO-graded recession and anteriorization. Four groups based on the inferior/temporal positions of reattachment of the IO (anterior border and posterior border together as one point) with respect to the IR lateral border as follows: 7.0/2.0 mm, 5.0/2.0 mm, 3.0/0.0 mm and 2.0/0.0 mm.

**Table 1 jcm-10-04433-t001:** Associated clinical factors and outcomes after inferior oblique muscle surgeries.

	IO Myectomy (*n* = 102)		IO-Graded Recession (*n* = 65)		# Changed Torsion	*p* Value *
OP	Pre-OP	Post-OP	Pre-OP	Post-OP		
Age range (y/o)	1.8–41		1.5–42			0.97
Head title (Degree)	12.5 ± 5.5	5.25 ± 3.5	13.5 ± 6.5	5.5 ± 4.0		0.94
Vertical deviation (Prism)	20 ± 4	6 ± 3	18 ± 6	6 ± 4		0.89
IOOA (Degree)	2.5 ± 1.5	1.25 ± 1.5	3.25 ± 1.25	1.5 ± 1.25	0.003	0.83
SOUA (Degree)	3.5 ± 2.5	1.5 ± 1.5	3.5 ± 3.5	1.5 ± 1.5	0.038	0.93
Excyclotorsion (Degree)	16.6 ± 6.4	5.1 ± 3.1	17.1 ± 5.2	5.5 ± 2.5	0.025	0.98

* *p* value: Compared the results of IO myectomy and IO-graded recession and anteriorization. # Regression model (*p* < 0.001).

**Table 2 jcm-10-04433-t002:** The preoperative and postoperative degrees of ocular torsion.

Groups	No.	Pre-OP (Degree)	Post-OP (Degree)	Net Change (Degree)	Reduction (%)	*p* Value
Overall	167	15.3 ± 6.4	5.3 ± 2.7	9.7 ±4.6	63.2	0.0007
Myectomy	102	16.6 ± 5.8	5.1 ± 3.1	10.7 ± 4.8	71.0	0.0007
IO graded recession	65	11.3 ± 3.4	4.2 ±3.3	8.1 ± 3.7	65.9	0.0009 (0.976 ^$^)
IO only	50	15.2 ± 6.7	5.6 ± 2.9	9.8 ± 3.9	65.4	0.98 *
Combine HR ^#^	117	14.3 ± 7.2	4.7 ± 2.9	9.8 ± 4.6	69.3	
Uni. SOP	45	15.8 ± 4.2	6.1 ± 2.5	10.7 ± 4.3	65.7	0.032 ^&^
Bil. SOP	122	12.1 ± 5.1	5.1 ± 3.2	8.2 ± 3.2	61.1	

# Combined HR: combined horizontal muscle operation. $: Compared with the effect of IO myectomy and IO-graded recession and anteriorization. *: Compared with the effect of IO only or in combination with HR. &: Compared with the effect in unilateral SOP and bilateral SOP.

**Table 3 jcm-10-04433-t003:** The effective components of full correction and partial correction.

	Full Correction	Partial Correction	*p* Value
Age (y/o)	12.32 ± 2.4	13.52 ± 3.1	0.982
Pre-op torsion (degree)	15.12 ± 3.2	16.26 ± 4.7	0.089
Net changes of torsion (degree)	10.18 ± 4.2	9.73 ± 3.5	0.064
Pre-op vertical deviation (prism)	12.23 ± 2.3	13.21 ± 3.8	0.25
Knapp’s Classification I–III (No)	111	21	0.044
Knapp’s Classification IV–VI (No)	18	12	0.0032 (*p* = 0.023) *

* Compare Knapp’s Classification I–III and Knapp’s Classification IV–VI.

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
