# Peer review of "The Effects of Modified Graded Recession, Anteriorization and Myectomy of Inferior Oblique Muscles on Superior Oblique Muscle Palsy"

_jcm, 2021, doi:10.3390/jcm10194433_

Round 1
Reviewer 1 Report
The current work needs further enhancements:
The statistical section is underdeveloped, and some critical information is lacking:
A-Line 157: The authors included 167 eyes. How did they choose this number? Did they try to carry out any sampling/experimental design before conducting the study? Determining the optimal sample size in the statistical analysis study could provide readers with the adequate number of participants needed to detect significant robust results! The GPower tool can do this calculation.
B- student t-test was used to compare the quantitative data. The student t-test is not appropriate for the authors' data for the following reasons:
The authors did not specify what variables were having a normal distribution.
Even if these variables show a normal distribution, no solid conclusions can be drawn because of the limited sample size. Thus it is highly recommended to use a non-parametric test such as the Mann-Whitney U test.
C- The authors have used a multivariate linear regression model and mentioned some variables such as age, etc.. This is not enough; more precision is needed; what variables are dependent, which are independent.
Lines 43, 45, 50, 51, etc.: Many key sentences throughout the text lack adequate reference(s). The authors need to add reference(s) at the end of every sentence.
Tables 1-3: The authors did not format the tables similarly in their manuscript. For example, in Tables 1 and 3, the significance column was put horizontally, whereas in table 2, it was vertically. It is always recommended to put the significance column vertically at the right end of the table. The authors are encouraged to re-format their tables with the help of a professional service.
Lines 62-67: Ethics statement and IRB approval; this section is repeated at the end of the manuscript in the sections Institutional Review Board Statement and Informed Consent Statement (lines 309-314). The authors are encouraged to omit it in the methods section.
Line 280: a limitation section is lacking. The authors can discuss the very small sample size, the retrospective nature of the study, etc…
Reviewer 2 Report
The manuscript entitled “The effects of modified graded recession, anteriorization and myectomy of inferior oblique muscles on superior oblique muscle palsy” aims to investigate the effect of inferior oblique (IO) operation (IO myectomy or graded recession and anteriorization) for unilateral and bilateral superior oblique muscle palsy (SOP). Their results showed that head tilt, vertical deviation, IO overaction, SO underaction degree and ocular torsion angle were all changed obviously but there was no statistically significance between these two procedures. Mean preoperative torsional angle was 15.3 ± 6.4 degree, which decreased to 5.3 ±2.7 degree after surgery. Preoperative torsional angle, IOOA and SOUA degree were all significantly affected postoperative torsional angle (p =0.025, 0.003 and 0.038). Horizontal rectus muscle and IO muscle operation did not interfere each other results (p= 0.98). Current manuscript is well written and interesting for readers. I think the authors should clarify more clearly about the selection of IO myectomy and graded recession for their patients. Were they selected randomly or based on what kind of clinical difference? I also suggest the authors can provide and compare a long term and short term results for their patients.Author Response
Please see the attachment.
